# Structural and Technological Characterization of Tropical Smallholder Farms of Dual-Purpose Cattle in Mexico

**DOI:** 10.3390/ani10010086

**Published:** 2020-01-05

**Authors:** Jaime Rangel, José Perea, Carmen De-Pablos-Heredero, José Antonio Espinosa-García, Paula Toro Mujica, Marisa Feijoo, Cecilio Barba, Antón García

**Affiliations:** 1Instituto Nacional de Investigaciones Forestales Agrícolas y Pecuarias INIFAP, Medellín de Bravo 94277, Mexico; jquintos@yahoo.com (J.R.); espinosa_j@hotmail.com (J.A.E.-G.); 2Departamento de Producción Animal, Universidad de Córdoba, Campus Universitario de Rabanales, 14071 Córdoba, Spain; pa2pemuj@uco.es (J.P.); cjbarba@uco.es (C.B.); pa1gamaa@uco.es (A.G.); 3ESIC Business & Marketing School and Departamento de Economía de la Empresa (Administración, Dirección y Organización), Economía Aplicada II y Fundamentos de Análisis Económico, Universidad Rey Juan Carlos, Paseo de los Artilleros, s/n, 28032 Madrid, Spain; 4Instituto de Ciencias Agronómicas y Veterinarias. Universidad de O’Higgins, San Fernando 3070002, Chile; paula.toro@uoh.cl; 5Departamento de Análisis Económico, Universidad de Zaragoza, Gran Vía 2, 5005 Zaragoza, Spain; mfeijoo@unizar.es

**Keywords:** ALC countries, smallholders, dual purpose, tropical zone, technological innovation

## Abstract

**Simply Summary:**

Dual-purpose cattle smallholder farms (DP) in the tropics suffer from sustainability and viability problems. Grouping small producers according to their structure and characterizing them technologically makes it possible to identify the problems aimed to guide development policies. A sample of 1475 farms located in the tropical area of Mexico was selected. Five groups of smallholders were identified applying multiple correspondence analysis (MCA). Results show that to achieve a sustainable improvement of the DP, a deep understanding of the system, the rational use of the endogenous resources, and implementation of low-cost technologies is necessary. Very small farms (Group 3) showed orientation to subsistence. They need to improve all the technological areas. Groups 1 and 2, covered a 46.5% of the farms; these ones presented a small-scale productive model and the improvements were mainly associated to the area of reproduction and genetics. Groups 4 and 5 (29.4% of the sample) were the biggest and more specialized farms. The improvements were linked to technological areas of reproduction, feeding, management, and animal health.

**Abstract:**

Dual-purpose cattle smallholder farms (DP) exhibit a critical economic situation. The objective of this research was building a typology for DP in tropical conditions and characterizing them technologically. This will help developing more effective public policies in DP farms located in tropical conditions. A sample of 1.475 farms located in the tropical area of Mexico was selected. The typology was built using multiple correspondence analysis (MCA). Subsequently, five groups were identified by a hierarchical cluster analysis with Ward’s method. Groups 1 and 2, covered a 46.5% of the farms; these ones presented a small-scale productive model with low levels of technological adoption, improvements were mainly associated to the area of reproduction and genetics. Very small farms (Group 3) showed orientation to subsistence. They need to improve all the technological areas. Groups 4 and 5 (29.4% of the sample) were the biggest and more specialized farms. Group four farms were located in dry tropics and showed the highest levels of technological adoption in the areas of reproduction, management, and feeding. These farms require improvement in the areas of reproduction, animal health, and feeding. Group 5 farms were located in the wet tropics and showed specialization in reproduction, genetics, and animal health areas. In this last group, it is necessary to improve management and feeding areas.

## 1. Introduction

Livestock production in Latin-America is characterized by a high variability of climate and agro-ecologic conditions. Therefore, there is a wide range of technical scenarios. United Nations for Food and Agriculture (FAO) and Pan American Dairy Federation (FEPALE) propose two types of organization for primary production in Latin American and Caribbean (ALC) [1], pastoralism and specialized production systems mainly in southern countries, and dual-purpose farms in the rest of ALC countries [2].

Dual-purpose (DP) cattle production systems have been traditionally preferred by family farms in the tropics due to their great flexibility, the adaptation to the climate conditions, and less capital investment and technical support required than specialized milk production systems [3]. Dual purpose cattle are usually raised in low-input systems based on natural resources, where the elements interact among themselves and with their natural context, providing smallholders with beef and milk from the same animal [4]. The objectives of the dual-purpose vary significantly according to the preferences of the farmer, weather, household consumption, local market, and the proportion of incomes generated from sales of meat and milk, which allows a wide variety of production models [5,6]. Dual-purpose constitutes a subsistence system and an important activity for the economic development of the smallholders in Latin America. It represents 70% of the total of livestock farmers from the tropical and subtropical regions. The predominant organizational model is the small-scale family farm; this means that, the farm is managed by the family, who also provides the main labor force [7]. The need to increase the information on their functioning, improving their low level of technology adoption and the analysis of their apparent resistance to adopt more innovative strategies are amongst the main challenges of the small scale systems. Higher levels of technology adoption are associated with an increased competitiveness, sustainability, and viability of farms [8,9,10].

The construction of typologies in tropical systems have a great value; contributing to DP knowledge and to the implementation of feasible improvement plans. A review of the main studies focused on the typologies of DP farms in ALC countries can be seen in Appendix A. However, they involve some methodological shortcomings such as: (a) In general, very unequal farms are compared (intensive with subsistence farms), resulting in predictable groups and classified as a supposed dual purpose [8]; (b) the farms are unequal regarding size and technologies, so, large, medium or small farms could prevail [6]; (c) in the principal components analysis (PCA), the number of variables is usually limited to the first two or three components [2,5]; (d) PCA or factorial analysis (FA) and cluster analysis are generally used, although there are other efficient methodological approaches with qualitative variables as multiple correspondence analysis (CMA) [11,12]; (e) the typologies are frequently elaborated at a local level, but typologies of country level or ecological zones are nonexistent [2,6,11,13].

Therefore, the objective of this study is to build a structural and technological characterization of DP farms through the development of a typology for smallholders in the tropical zone of Mexico. This research will permit first, to analyze the great variability among smallholders and second, to build groups of DP at a global level. The proposed groups could then be used to design strategic public and private policies for technological development of the sector.

## 2. Materials and Methods

### 2.1. Study Area

This study was conducted in the Central and Southeastern Coastal region of Mexico. The region under study presents a tropical climate with high temperatures; it is divided into two ecological zones: Dry tropic (DT) with an extended drought period that varies between five and nine months and less than 1200 mm of rainfall, and wet tropic (WT) with rainfalls over 1200 mm of rainfall distributed throughout the year [14].

### 2.2. Data Collection

The data of dual-purpose bovine farms come from the research project “Adoption and evaluation of the impact of technology implemented in dual purpose bovine systems of México”, approved by the Collegiate Group of the National Center for Disciplinary Research in Physiology and Animal Improvement of INIFIAP SIGI 21541832011. This project was approved in 14 March 2011 and was performed from 1 April 2013 to 30 April 2016. Data were collected in 2015 by direct surveys done to smallholders of dual purpose. The farms were located at municipalities with high levels of social marginalization of Mexican tropics [1,6]. The sample was chosen among 3285 commercial farms of DP receiving technical advice on feeding and economical management (Livestock Technical Assistance Program, SAGARPA, 2019). Sampled farms were selected (n = 1475), after eliminating the extreme values of the structural features (e.g., farms without structure or organized productive activity), according to the availability and reliability of the records and technical criteria of advisors. Only farms that sold animals the previous years were selected. Data from farms were collected by filling out the respective questionnaires jointly by the stakeholder and the technical advisers.

Thirty-one variables were selected to build the typology, thirteen of these were demographic and structural variables: Province (9), agroecological zone (dry or wet tropic), grazing surface (ha), animal units (UA), herd size (heads), heads in production (cows), stocking rate (UA/ha), land ownership (private or communal “Ejido”), grazing planting (0 = No, 1 = Yes), grazing crop residues (0 = No, 1 = Yes), silage (0 = No, 1 = Yes), hay (0 = No, 1 = Yes), green fodder (0 = No, 1 = Yes). Ten out of the 31 were productive variables: Milk yield (L/farm/year), milk per cow (L/cow/year), calves (heads), unproductive animals (cows that do not produce milk or had a calf for more than a year, heads), cheese yield (kg), milk yield per worker (L/W), milk production per ha (L/ha), milk sold (0 = No, 1 = Yes), cattle sold (0 = No, 1 = Yes), dairy products sold (0 = No, 1 = Yes). Finally, eight variables were related to social aspects: Age (years), economic dependents (n°), employees or work units (W), gender (male, female), farmer’s education (three levels), farm’s income (%), marginalization level by the State of Mexico in 2013 (five categories from very high to very low) [14], commercial channel (direct, short, and long; without intermediaries, with a single or more than one, respectively).

### 2.3. Livestock Innovation Level

The technological level of the farms was evaluated according to the methodological approach used by [5,15,16]. The selection of innovations and its grouping into technological areas took place according to a qualitative, consensus, and participatory methodology described by [17]. Briefly, the methodology consists of two steps. In the first step, relevant technologies and best practices were identified and selected. In a second step, technologies were grouped into technological areas. Forty-five technologies were identified and distributed as follows: Eight in the management area, fourteen in feeding area, nine in genetics area, seven in reproduction area, and seven in the animal health area. A technological innovation index was calculated per each area. It was based on the proportion of technologies implemented overall technologies identified for each area (Appendix A).

### 2.4. Statistical Analyses

The classification of farms was made by a multivariate approach based on quantitative and qualitative variables. The 12 selected variables to perform multiple correspondence analysis (MCA) are shown in Table 1. The MCA reduces the dimensions of contingency tables and provides a graphical representation of the information contained in these tables in a new space of independent factors where the similar categories of different variables appear closer [13,14,18]. Quantitative variables were transformed into classes using the quantile position with respect to the median. A stepwise procedure was carried out for checking for at least 20 cases in each of the categories. The appropriateness of incorporating the variables in the MCA was assessed by the chi-square test between pairs of variables. Those variables that showed few associations were removed. To select the proper number of dimensions, those with eigenvalues greater than the value of the mean were selected [19], and the proper alpha Cronbach index [20] was calculated. The MCA allowed detecting the factors (dimensions) that best characterize the farms. These factors can be directly used in the subsequent analysis without the need to be standardized.

A hierarchical cluster analysis was done with the dimensions showing the greatest variance generated by the MCA. Cluster analysis allowed grouping the farms that were similar between them (minor within-group variance) and different to the others (greater variance between groups). The groupings were done based on Ward’s method, using the Euclidean, squared Euclidean, and Manhattan distances [21]. Finally, by making use of benchmarking techniques, a comparison between farms with high technological (25% of the farms) and low technological level (75%) was done for each area and identified group [22]. All data were analyzed using statistical software SPSS version 19 [23].

## 3. Results

### 3.1. Characteristics of Smallholder Dual-Purpose Cattle Farms

Descriptive statistics are shown in Table 2. The average farm had 29.9 UA of herd size in 35.6 ha and 1.2 UA/ha of stocking rate. Animal feeding was based on grazing native pastures (*Paspalum*, *Panicum*, *Bouteloua*, etc.) and grazing crop residues (52.7%); 39% of the farms feeding with grazing planting, silages were utilized in 22.5%, 50.4% hay, and 30.3% green fodder. In [16,22], the dual-purpose system in Mexico is widely described.

The technological level of adoption was low, only 47.0% of the technologies potentially available were used. Animal health was the area with higher technological adoption, followed by the management and genetics. The less used technologies were feeding and reproduction. While the structural characteristics of the farms were quite heterogeneous, the technological levels were quite homogeneous (Table 2).

### 3.2. Smallholder’s Dual-Purpose System Typology

The MCA analysis yielded only five dimensions, as these were the only ones with eigenvalues exceeding the value of their mean. These five dimensions accounted for 71.3% of the inertia and the average value of Cronbach’s alpha was 0.726, which was considered acceptable [19,20]. The cluster analysis with the most significant results was the solution of five groups with Ward’s method, based on the Euclidean distances (Figure 1).

The farms of Group 1 (26% of the total) were distributed into both agro-climatic zones (DT and WT) and they were characterized by their low stocking rate and intermediate production and size. There were no significant differences in the grazing crop residues either in the hay production (Table 3 and Table 4). The technological level was intermediate, while the lowest values were shown in genetics and reproduction areas (Table 5).

Group 2 was composed by 20.5% of the sample (n = 1475), mainly characterized by their low stoking rate and intermediate size both in terms of animals and surface. Most parts of the farms that are in this group are situated in the dry tropic. Most of the farms produced hay and used grazing crop residues as feed. The yield per cow was the lowest of all groups, while the yield per ha was intermediate (Table 3 and Table 4). These farms presented a higher technological level in the areas of management and feeding, and a medium-low level in the genetics and reproduction areas (Table 5).

Group 3 concentrated 24% of the farms, which were mainly characterized by small size and extremely low annual productivity. Likewise, labor productivity was the lowest of all groups. The marginalization level of these farms was from high to very high in most cases. Around a third of farmers indicated that their income depended on 50% in the DP cattle (Table 3). No differences were found in the use of grazing crop residues and hay (Table 4). Livestock was oriented to familiar subsistence; the few surpluses were sold in local markets. This group presented a low level of technological adoption, also in each analyzed area (Table 5).

Groups 4 and 5 were composed by large farms with a business orientation. Group 4 was mainly located in DT and it was composed by the farms with the largest size and production. Most farms used grazing crop residues. It was the group with the highest labor force productivity. Nevertheless, the yield per cow was intermediate and per ha was low (Table 3 and Table 4). These farms were oriented to cheese, milk, and beef production. They exhibited a higher technological development in management and genetics areas. Although their general technological level was very low in reproduction, and low in animal health and feeding (Table 5). Group 5 was the smallest group and it was located mainly in WT zone. Most farms produced hay and were using the surplus forages from the rainy season. They stood out by having the highest stocking rate and a low number of unproductive animals. These farms were oriented to beef-milk-cheese production, although with lower yields and land than Group 4 (Table 3 and Table 4). This group was differentiated by the higher technological level in the areas of genetics, reproduction, and animal health. The technological level in management and animal feeding was very low (Table 5).

## 4. Discussion

A typology of smallholders (less than 50 cows), of dual purpose cattle (meet-milk) at a country level in Mexico was built using a survey and technological characterization developed by [16,17]. The dual-purpose system can be found in the Central and South American tropical areas, from Mexico to northern Brazil, including Colombia, Venezuela, Ecuador, and Peru [1]. In this research the typology has been built at a country level taking into consideration most of the variability in the tropic; 1475 smallholders, two ecological zones (Dry and Wet), 9 states, and 62 million ha. were considered. Therefore, this research contributes to country level and can be generalized to other similar contexts. Multiple correspondence analysis and cluster analysis were used [11,12,21]. Five identified groups were characterized according to adopted technologies; 45 technologies grouped in the areas of management, feeding, genetics, reproduction, and animal health. The range of technological variation was low (from 43.7% to 50.2%) although differences among groups were identified and the technologies adopted in each cluster differed. The typology was similar to other farms found in different states in Mexico with high levels of marginalization, low dependence on external inputs, and very low technological level [24,25,26,27,28,29] (Appendix A), but with strong contrasts in regard to production scale, and gross incomes. In contrast, typologies from developed countries showed a medium-high technological level, high dependence on external incomes, and low marginalization level [11,12,17].

The proposed typology and their technological characterization are key tools to design strategic policies [6,13]. Since an increase in technological adoption is associated with higher levels of competitiveness and sustainability [8,9]. In a second researching stage; what should the priority technologies be in each group for the design and development of an effective public policy? We tried to approach the challenge by reviewing the identified dual-purpose technologies in ALC countries (Appendix A), previous studies [15,16,22] and the results of this research (Table 3, Table 4 and Table 5, and Figure 2). We propose the implementation of technologies presenting low adoption level, where the improvement turns into a strong impact, according to the curve of diminishing returns of innovation. In addition, we also select low cost technologies, appropriate to the context, and easy to implement by producers [8,15,25,30].

For each group, the less developed technological areas were quantified by benchmarking [22]. The development of specific strategic smallholders’ policies for five groups identified in the typology will allow reaching higher levels of technological adoption. Groups 1 and 2, responded to a small-scale productive model, addressed to sell surpluses into short and direct commercialization channels, generally addressed to a local market. Strategic policies for smallholders for these groups could be oriented to reproduction and genetics areas, on technologies such as semen evaluation, evaluation in bulls, pregnancy diagnosis, oestrus detection, female evaluation, breeding policy, tested bulls, calves’ selection, and female selection.

Groups 4 and 5 presented the largest size, with a business orientation, and their production was addressed to short and long commercialization channels. In Group 4 the technological improvement priorities should be oriented to the area of reproduction with the measures indicated above. However, the farms require enhancing the areas of animal health and feeding. Improving technologies such as health planning, parasite diagnosis, mastitis, and sanitary milking would be necessary for animal health [12,17,22,26]. In the animal feeding area, it will be necessary to increase the use of green fodder, use of agro-industrial by-products, fodder preserved as silage or hay, processed feeding, salt and vitamins blocks, molasses/urea, grains and oilseeds, and mineral blocks [27,30,31]. Group 5, farms located in wet tropic, were oriented to beef-milk, and their main innovations were composed by forages and hay reserve bales. The technological areas of genetics, reproduction, and animal health were the most implemented. The management and feeding areas need to be supported in order to develop adequate technologies for wet zones as record systems, grazing planting, crop residues, and adopting the mechanical milking [2,7,8,13,16,28].

Moreover, previous research associates directly the size, agro climatic zone, and sustainability of the farm with its technological level [7,8]. They indicate that producers should increase dimension at a great extent, otherwise a significant number of smallholders will be expelled from the system and will have to become laborers in their community or migrate to urban areas to find employment [32]. The increase of scale and intensification is an important aspect, although it is not enough for reaching a sustainable development in the tropics [2,7,10,11,17]. Our results showed that size was not associated to technological level. Group 2 showed higher technological level than Group 1 (small scale model). Furthermore, Group 4 showed higher technological level than Group 5 (medium scale model). Nor was the technological level associated to ecological zone. Possibly, the technological level was associated to the productive level in each cluster [5,8,12,13].

In this way, Hernández-Castellano et al. [13] indicated the paradigm shift on the old fashion question of “how much milk is produced” has been replaced by “how milk is produced”. This implies an evolution process, from the productive intensification towards sustainable productivity that includes aspects related to food sovereignty, environmental health, and agrosystem [31]. The process is focused on the use of endogenous resources and minimizes the use of external inputs. Therefore, the strategy presents a sustainable orientation and implies a deep knowledge of the potential of the territorial resources as residues, grazing, planting, silage with deep inclusion of tropical by-products [16,28] and, the technologies [7,27]. In order to increase efficiency, technology has been related to development [25,29]. In addition, from a circular and blue economy approach, it is necessary to seek equilibrium between the utilized breeds and the available food resources related to nutritional and reproductive management [8,29].

The development of farm typologies constitutes a tool to identify structural features of production systems, generate a framework within which policies may address the needs of specific farm categories, and identify farms with a need or potential to adopt new technologies [18,29]. The findings of the study also indicate that farms belonging to different typologies, may need different (advisory) approaches to achieve the goal of increasing their technological level [12]. The research is exploratory and further research evaluating the incidence of technologies in the results through logistic regression, structural equation model (SEM); identifying the causes of inefficiencies with DEA, etc. [6,7,9,10] should be done.

## 5. Conclusions

MCA and cluster analysis have been shown to be a useful tool for the identification of productive systems in double purpose in Mexican tropical areas. Five groups of dual-purpose cattle smallholder farms (DP) have been identified. Group 3 was characterized by a very small size, oriented to familiar subsistence, and presented a low level of total technological adoption. Groups 1 and 2 responded to a small-scale productive model and their strategic policies should be oriented to reproduction and genetics area. Groups 4 and 5 presented the larger size, with a business orientation. Group 4 is composed of farms located in dry tropics, requiring improvements in the areas of reproduction, animal health, and feeding. In Group 5, composed of farms located in the wet tropics, it would be necessary to improve management and feeding areas.

## Figures and Tables

**Figure 1 animals-10-00086-f001:**
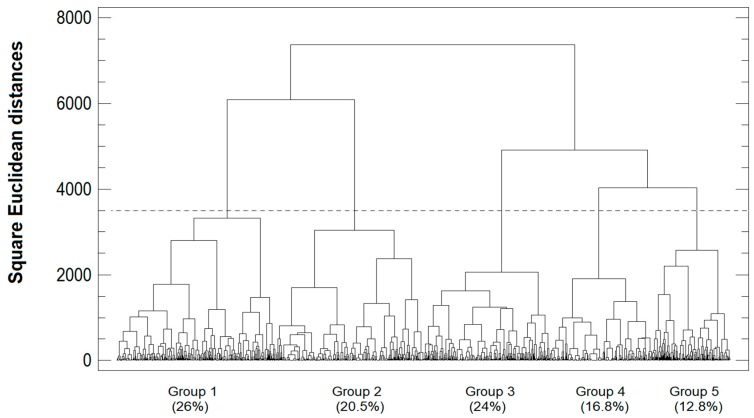
Dendrogram classification of dual-purpose farms in Mexico (farms, %).

**Figure 2 animals-10-00086-f002:**
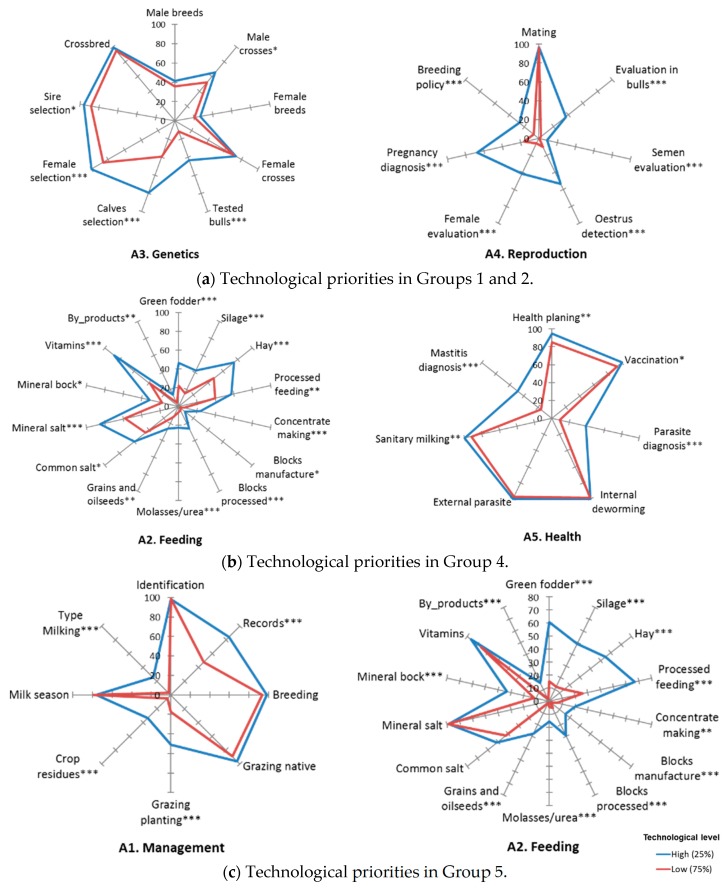
Technological priorities in each group of smallholder dual-purpose farms. *** Significant differences between farms with high (25%) and low (75%) technological level (* *p* < 0.05, ** *p* < 0.01, *** *p* < 0.001).

**Table 1 animals-10-00086-t001:** Structural indicators and their classes used in the multiple correspondence analysis of smallholder’s dual-purpose system (n = 1475).

Variable	Class	n
Provinces or State		
Campeche	1	45
Chiapas	2	440
Colima	3	52
Michoacán	4	206
Morelos	5	36
Oaxaca	6	36
Quintana Roo	7	55
Sinaloa	8	560
Tabasco	9	45
Grazing surface, ha		
I	≤13	371
II	14–22	376
III	23–42	361
IV	≥43	367
Total animal unit, UA		
I	≤14.9	372
II	15–24.5	369
III	26.6–39.6	367
IV	≥39.7	367
Milk per cow, L/cow/year		
I	≤456.2	372
II	456.2–774.2	366
III	774.2–1200.0	378
IV	≥1200.0	359
Milk production per ha, L/ha		
I	≤15.4	369
II	15.4–38.0	369
III	38.0–90.0	369
IV	≥90.0	368
Dimension cows in production		
Very small	<9 cows	410
Small	<20 cows	538
Medium	20–50 cows	527
Total milk yield, L/farm/year		
I	≤5400	377
II	5401–11,250	361
III	11,251–18,100	369
IV	≥18,101	368
Agroecological zone		
Dry tropic	1	1109
Wet tropic	2	366
Herd size, heads		
I	≤19	369
II	20–33	379
III	34–53	367
IV	≥54	360
Milk yield per worker, L/worker		
I	≤3000	388
II	3001–6600	352
III	6601–12,960	368
IV	≥12,961	367
Total production cows, heads		
I	≤9	410
II	10–14	333
III	15–23	377
IV	≥24	355
Grazing crops residues		
No	0	698
Yes	1	777

**Table 2 animals-10-00086-t002:** Structural and technological characteristics of smallholder dual-purpose farms (n = 1475).

Item	Mean	Median	SD ^1^	CV ^2^	Min ^3^	Max ^4^
Grazing surface, ha	35.6	22.0	48.8	137.0	0.0	600.0
Total of animal unit, UA	29.9	24.5	20.0	67.0	0.0	133.6
Herd size, n° cattle	39.8	33.0	26.9	67.8	6.0	183.0
Stocking rate, UA/ha	1.2	1.0	0.7	61.7	0.0	6.2
Milk production, L/year	14,688	12,000	15,241	103.8	0.0	150,000
Milk per cow, L/cow/year	891.2	810.4	603.5	67.7	0.0	3842.1
Calves sold, n° calves	6.1	3.0	6.8	110.9	1.0	55.0
Unproductive animals, heads	3.1	0.0	5.5	180.8	0.0	90.0
Cheese yield, kg/farm/year	304.3	0.0	850.5	279.5	0.0	10,000.0
Milk yield per worker, L/worker	9882	7200.0	11,597	117.4	0.0	150,000.0
Milk production per ha, L/ha	86.6	39.9	158.6	183.2	0.0	2.140.0
Stakeholders age, year	51.5	51.0	13.9	26.9	18.0	85.0
Economics dependents, n	3.0	3.0	2.0	67.2	1.0	36.0
Employments, workers	1.8	1.0	1.7	96.8	1.0	19.0
Total production cows, heads	17.1	14.0	11.1	64,9	6.0	50.0
Total technological level (%)	47.0	48.9	11.6	24.7	0.0	100.0
Management (%)	61.0	50.0	15.5	25.5	0.0	100.0
Feeding (%)	27.6	21.4	16.5	59.8	0.0	100.0
Genetics (%)	59.8	55.6	16.0	26.8	0.0	100.0
Reproduction (%)	27.2	28.6	20.2	74.2	0.0	100.0
Animal health (%)	72.9	71.4	15.9	21.9	0.0	100.0

^1^ Standard deviation, ^2^ Coefficient of variation, ^3^ Minimum, ^4^ Maximum.

**Table 3 animals-10-00086-t003:** Structural characteristics and performance for each group of smallholder dual-purpose farms.

Cluster	Group 1	Group 2	Group 3	Group 4	Group 5
Farms, %	26.0	20.5	24.0	16.7	12.7
Grazing surface, ha	27.2 ± 38.7 ^b^	37.4 ± 39.5 ^c^	16.5 ± 33.1 ^a^	74.6 ± 77.4 ^d^	34.5 ± 19.8 ^c^
Total of animal unit, UA	19.3 ± 4.0 ^b^	32.1 ± 6.5 ^c^	10.9 ± 4.8 ^a^	60.2 ± 17.5 ^e^	43.7 ± 16.5 ^d^
Herd size, n° cattle	25.5 ± 6.3 ^b^	43.1 ± 9.4 ^c^	14.3 ± 7.2 ^a^	80.7 ± 23.4 ^e^	57.6 ± 21.9 ^d^
Stocking rate, UA/ha	1.1 ± 0.6 ^a^	1.2 ± 0.7 ^a^	1.1 ± 0.8 ^a^	1.2 ± 0.8 ^a^	1.4 ± 0.6 ^b^
Milk production, L/year	11,229 ± 6824 ^b^	15,346 ± 9753 ^c^	6168 ± 5771 ^a^	27,968 ± 22,724 ^d^	19,267 ± 20,576 ^c^
Milk per cow, L/cow/year	987.7 ± 591.7 ^b^	831.4 ± 500 ^a^	937.3 ± 678 ^b^	876.8 ± 609 ^a,b^	722.9 ± 579 ^a^
Calves sold, n° calves	4.9 ± 5.8 ^b^	6.6 ± 5.9 ^c^	3.6 ± 4.1 ^a^	10.8 ± 8.8 ^d^	6.2 ± 7.6 ^b,c^
Unproductive animals, heads	2.5 ± 4.5 ^a^	2.7 ± 3.6 ^a^	2.5 ± 6.8 ^a^	5 ± 6.2 ^b^	3.2 ± 5.9 ^a^
Cheese yield, kg/farm/year	245.2 ± 733.7 ^a,b^	317.5 ± 689.6 ^b,c^	130.4 ± 350.5 ^a^	539.8 ± 1184.8 ^c^	421.5 ± 1266.6 ^b,c^
Milk yield per worker, L/worker	8572 ± 6827 ^b^	10,559 ± 9430 ^c^	4581 ± 4613 ^a^	17,144 ± 19,649 ^d^	11,899 ± 11,921 ^c^
Milk production per ha, L/ha	107.8 ± 186.8 ^c^	72.7 ± 119.7 ^b^	132.2 ± 194.8 ^c^	44.1 ± 120.2 ^a,b^	35.8 ± 56.8 ^a^
Stakeholders age, year	51.1 ± 14.5 ^a,b^	53.4 ± 13.4 ^b^	49.8 ± 13.9 ^a^	53.1 ± 13.2 ^b^	50.7 ± 13.8 ^a,b^
Economic dependents, n	2.9 ± 1.8	3.1 ± 1.9	2.8 ± 1.8	3.4 ± 2.8	2.9 ± 1.6
Employees, workers	1.5 ± 1.1 ^a^	1.8 ± 1.5 ^b^	1.5 ± 1.5 ^a^	2.5 ± 2.4 ^c^	2.1 ± 2.2 ^b,c^
Total production cows, heads	11.8 ± 2.9 ^b^	18.9 ± 5.2 ^c^	6 ± 2.3 ^a^	32.4 ± 9.2 ^e^	25.6 ± 10.3 ^d^

^a–e^ Means within a row with different superscripts differ significantly (*p* < 0.05).

**Table 4 animals-10-00086-t004:** Characterization qualitative variables according to groups obtained for smallholder dual-purpose farms.

Item	Class	Overall ^1^	Group ^1^					*p*
			1	2	3	4	5	
Grazing crops residues	No	47.3	43.9	32.3 *	48.3	39.3 *	87.2 *	0.000
	Yes	52.7	56.1	67.7 *	51.7	60.7 *	12.8 *	
Hay	No	49.6	49.9	34.7 *	53.1	43.3 *	74.5 *	0.000
	Yes	50.4	50.1	65.3 *	46.9	56.7 *	25.5 *	
Ecological zone	Dry	75.2	73.1	96.4 *	78	98.4 *	9.6 *	0.000
	Wet	24.8	26.9	3.6 *	22	1.6 *	90.4 *	

^1^ Values are in percentages; * Significant differences between rows within the same group (Chi square test).

**Table 5 animals-10-00086-t005:** Technological characterization for each group of smallholder dual-purpose farms.

Technological Area	Group				
	1	2	3	4	5
Mean level, %	47.4 ± 10.6 ^b,c^	48.4 ± 12.3 ^c^	43.7 ± 10.0 ^a^	46.5 ± 11.9 ^b^	50.2 ± 12.5 ^c^
Management, %	60.7 ± 17.1 ^a,b,c^	64.9 ± 14.4 ^c^	57.4 ± 15.9 ^a^	64 ± 13.9 ^c^	58.4 ± 13.3 ^a,b^
Feeding, %	27.4 ± 16.1 ^a,b,c^	30.1 ± 16.4 ^c^	25 ± 15.4 ^a^	29.6 ± 15 ^b,c^	26.5 ± 19.5 ^a,b^
Genetics, %	59.2 ± 14.7 ^a^	61 ± 15.4 ^b^	54.7 ± 17.6 ^a^	62.3 ± 14.7 ^b^	65 ± 15.5 ^b^
Reproduction, %	27 ± 19.6 ^a^	26.7 ± 20.9 ^a^	27.2 ± 19.6 ^a^	23.3 ± 16.3 ^a^	33.7 ± 24.3 ^b^
Animal Health, %	72.5 ± 16.0 ^a,b,c^	73.2 ± 14.6 ^a,b,c^	70.2 ± 17.2 ^a^	74.3 ± 14.1 ^b,c^	76.1 ± 16.8 ^c^

^a–c^ Means within a row with different superscripts differ significantly (*p* < 0.05).

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
