# Peer review of "Structural and Technological Characterization of Tropical Smallholder Farms of Dual-Purpose Cattle in Mexico"

_animals, 2020, doi:10.3390/ani10010086_

Round 1
Reviewer 1 Report
Ethics approval is still missing, according to Animals Journal - Instructions for authors:
According to point 23 of this declaration, an approval from an ethics committee should have been obtained before undertaking the research. At a minimum, a statement including the project identification code, date of approval, and name of the ethics committee or institutional review board should be cited in the Methods Section of the article.
https://www.mdpi.com/journal/animals/instructions#ethics
The discussion section doesn't provide a strong justification of how the knowledge acquired through this study can benefit farmers, policy makers or animals.
Also, there is still no strong argument on how this research study contributes to the local and how this can be generalized to other or similar contexts.
Author Response
Many thanks for your comments. According to them, we have done the following changes to last version of the manuscript:
According to point 23 of this declaration, an approval from an ethics committee should have been obtained before undertaking the research. At a minimum, a statement including the project identification code, date of approval, and name of the ethics committee or institutional review board should be cited in the Methods Section of the article.
https://www.mdpi.com/journal/animals/instructions#ethics
The data of dual-purpose bovine farms come from the Research Project “Adoption and evaluation of the impact of technology implemented in dual purpose bovine systems of México”, approved by the Collegiate Group of the National Center for Disciplinary Research in Physiology and Animal Improvement of INIFIAP SIGI 21541832011. This project was approved in March, 14th, 2011, and was performed from 1st April 2013 to 30th April 2016.
A sentence with the required information from reviewer has been included in methodology.
Besides we include the letter for the journal with the authorization for the Project.
The article has been built in the framework of a Research Project authorised and funded by INIFAP, Mexico, and therefore, it meets the relevant ethical requirements: farmers have participated voluntarily given their consent to use data for the research (improvements proposal, publishing of results, etc.), data have been treated confidentially (data coming from farms and farmers were coded and anonymized), the information has been provided in an aggregated way (never on an individual basis for each farmer or farm).
The discussion section doesn't provide a strong justification of how the knowledge acquired through this study can benefit farmers, policy makers or animals.
According to reviewer’s comments, the following paragraphs have been included in the discussion:
The development of farm typologies constitutes a tool to identify structural features of production systems, generate a framework within which policies may address the needs of specific farm categories, and identify farms with a need or potential to adopt new technologies [18, 29]. The findings of the study also indicate that farms belonging to different typologies, may need different (advisory) approaches to achieve the goal of increasing their technological level [12].
What should be the priority technologies in each group for the design and development of an effective public policy? We tried to answer the question by reviewing the identified DP technologies in ALC countries (appendix A, Table A1), previous studies [15, 16, 22] and the results of this research (Tables 3, 4, 5, and Figure 2). The answer is technologies presenting low adoption level, where the improvement turns into a strong impact, according to the curve of diminishing returns of innovation. Besides, low cost technologies have been selected, appropriate and easy to implement by producers [8, 15, 25, 30].
For each group, the less developed technological areas were quantified by benchmarking [22]. Figure 2 shows the technologies responsible for the differences (P<0.001). The development of specific strategic smallholders’ policies for five groups identified in the typology will allow reaching higher levels of technological adoption. Groups 1 and 2, responded to a small-scale productive model, addressed to sell surpluses into short and direct commercialization channels, generally addressed to a local market. Strategic policies for smallholders for these groups could be oriented to reproduction and genetics areas (Figure 2a), on technologies such as semen evaluation, evaluation in bulls, pregnancy diagnosis, oestrus detection, female evaluation, breeding policy, tested bulls, calves’ selection, female selection (P< 0.001).
Also, there is still no strong argument on how this research study contributes to the local and how this can be generalized to other or similar contexts.
A typology of smallholders (less than 50 cows), of dual purpose cattle (meet-milk) at a country level in Mexico has been built. Dual purpose system is located mainly in the Central and South American tropical areas, from Mexico to northern Brazil, including Colombia, Venezuela, Ecuador and Peru [1]. In this research the typology has been built at a country level taking into consideration most of the variability in the tropic. 1475 smallholders, two ecological zones (Dry and Wet), 9 states and 62 million ha. were considered. So, this research contributes to country level and can be generalized to other or similar contexts.

Reviewer 2 Report
A revision of the manuscript was done, according to my suggestions (and to other two reviewer's advice). I believe that the manuscript has been improved.
I recommend to check the sentence at line 61-65 (there is a repeated phrase).
I asked authors to include in table 2 the Median and the Range values (minimum and maximum value); however, they included the Median and only the Maximum value. I suggest to modify it by adding the minimum value.
Author Response
Many thanks for your comments. According to them, we have done the following changes to last version of the manuscript:
I recommend to check the sentence at line 61-65 (there is a repeated phrase).
The sentence has been modified according to this suggestion.
I asked authors to include in table 2 the Median and the Range values (minimum and maximum value); however, they included the Median and only the Maximum value. I suggest to modify it by adding the minimum value.
The table 2 has been modified according to the suggestion (maximum and minimum value).

Reviewer 3 Report
The paper has been improved.
Author Response
Many thanks for your comments.

Round 2
Reviewer 1 Report
Dear authors, many thanks for the answers provided. Please find attached a PDF with my final comments on the manuscript.

Author Response
Thanks for the revision. Thanks for the time and effort dedicated to help us improve this manuscript.
The proposed modifications have been read and the required changes have been done.
Here, there is a summary of the modifications done:
Line 54: It has been modified
Line 58: It has been modified
Line 62: It has been modified
Line 72: Viability of farms, in general.
Line 81: It has been modified
Line 87-91: it has been modified
2.2. Data collection. It has been modified in deep according to each comment. We think now the manuscript is more clear and understandable now.
“These two paragraphs are about data handling and not data collection. Please add a sub heading accordingly to what is described here.”
The paragraph 2.3. Livestock innovation level has been added.
“Is this the reference? What is described here matches very well this other manuscript:
https://www.mdpi.com/1999-5903/8/2/25 Rivas et al is in sheep in Spain, nothing to do with dual purpose and LAC.”
This reference has been added:
García, A.; Rivas, J.; Rangel, J.; Espinosa, J.A.; Barba, C.; De-Pablos-Heredero, C. A methodological approach to evaluate livestock innovations on small-scale farms in developing countries. Futur. Internet. 2016, 8, 25. https://doi.org/10.3390/fi8020025.
The rest of comments have already been considered.
Results
“could you provide the reader with information on how you know this? I could not find any of these on the tables below. Do you have %? or how many of the farmers interviewed provided this information?”
The percentages have been included. Besides “In Rangel et al. [16, 22] dual-purpose system in Mexico is widely described”.
“Characterization of qualitative variables according to groups obtained for Smallholder Dual Purpose farms.
Please review and edit all the titles of tables and figures accordingly. Reader must be able to understand what the table is about without going back to the text”
All the tables and figures titles have been modified according to recommendations.
The rest of modifications have been done. They are in blue colour in the new version of manuscript.
Discussion
According to recommendations, the discussion has been restructured and the indicated paragraphs have been re-written.

This manuscript is a resubmission of an earlier submission. The following is a list of the peer review reports and author responses from that submission.
Round 1
Reviewer 1 Report
This paper provides information about characteristics of dual purpose farms in Mexico.
Introduction:
L52 - FAO-FEPALE needs to be define and a reference provided for this sentence
L60 - This reference does not match with the sentence here. In addition, the definition used seems to talk about a silvopartoril system and not only about a dual purpose system.
Materials & Methods: Before the study was carried out, was the design approved by an ethics committee? and did you inform the participants that they were part of a research study?
L95-L103 - Did the farms were selected because they sold animals the previous year? why did you decide to use this selection method? Did all the selected farms participated? Did farmers had the right to decline to not participate? if not, why not?
L115-120 Explain briefly how the technological level was evaluated.
L147 - what is an unproductive animal? calves have already been mentioned? sick animals? bulls?
Discussion
It describes findings but does not provide a strong argument for the study in the first place.
Reviewer 2 Report
I think that the aim of the present investigation is very interesting.
I recommend minor revision of the manuscript.
As you can see from my comments below, I find that the manuscript need some care, so I would encourage the authors to give it a self re-read and revision to improve it.
Below, some overall, yet detailed notes.
Language in general: The manuscript is well written; however, some sentences need to be revised by the authors. The authors use both present and past simple. Generally, the methodology and the findings of the paper should be in past tense, while the present tense should be used to discuss the results, for conclusions and implications, or to refer to tables and figures.
Simply Summary
L.24: should it be “multiple correspondence” rather than “multicorrespomdece“?
Statistical analysis
I have no specific comments for this section, since I am not a statistician, but my impression is that the chosen methods undoubtedly are appropriate for this type of data.
Results
Line 170-219: As above specified, the authors use the past simple for presenting results of Group 1, 4 and 5; then they use present and past tense for group 2 and 3, interchangeably. Please, use the past tense.
Line 188: should it be “Table 3 and 4” rather than “Table 4 and 5 “?
Line 192-199: please, add a reference to “table 3 and 4” in the text, as for the other groups
Line 204: should it be “876.8” rather than “ 976.8 “?
Line 212: should it be “low” rather than “the lowest “?
Line 213: should it be “421.5” rather than “21.5 “?
Line 213-214: This sentence is confusing. Please revise it.
Table 2: I think that authors could include the median value (and the range), as they could better characterize data.
Table 3 and table 5:
Please, check all the values have one decimal point.
Line 178: should it be “a,b,c,d,e” rather than “a,b,c “?
Table 4:
Please, delete the black line under “Grazing crops residues” and “No”.
Should “2” be replaced with “*”?
Discussion
Line 238-239: should you specify Figure 2b refer to Group 4?
Line 243: should it be “salt and vitamins” rather than “blocks “?
Line 246-247: Please, move “Figure 2c” after “Management and feeding areas…”.
Conclusions
Line 277: should it be “Group 5” rather than “Group five”?
Reviewer 3 Report
General comment.
The manuscript deals with a survey on dual purpose cattle farms in Mexico. The topic of the paper may have some interest, even though it is focused on a local situation. However the paper presents some shortcomings that preclude its acceptance in the present form.
The paper is excessively long for its real informative content. In the result section, the same data are presented both in table and listed in the text. On the other hand, some methodological aspects that could help the reader (for example what is intended for “genetic area” or “Reproduction area”) are neglected. Also in the discussion section, speculation are rather poor and the subsequent conclusions does not add, in my opinion, any point of interest or novelty in the field of livestock farming and management in general and, I think, also in the specific conditions examined in the paper.
|
Lines |
Comment |
|
51 |
I would suggest to change “takes place under variable” with “is characterized by a high variability of “ |
|
77-78 |
What do authors intend with the statement “… in the principal components analysis (PCA), the number of variables is limited…”? |
|
106 |
What is “ total herds (herds)2? Is the number of animals farmed in the herd? I would put “herd size” |
|
116 |
“…used by [7, 12, 13]”…. |
|
116 |
“..forty – five..”. it is a large number for the levels of a classification factor. Authors should add detail on this point. |
|
169-219 |
In this section are reported the same data shown in table 2. Please avoid to double-reporting data. This section should be greatly shortened. |